



# Lidar studies on microphysical influences on the structure and lifetime of tropical cirrus clouds

G S Motty[1*], Malladi Satyanarayana[1, 2], G S Jayeshlal[1], and V P Mahadevan Pillai[1]

[1.] Department of Optoelectronics, University of Kerala, Kariavattom, Trivandrum-695 581, Kerala, India

[2.] Geethanjali College of Engineering & Technology, Cheeryal (V), Keesara (M), R.R.Dist. Hyderabad-501 301, India

*Correspondence to*: G S Motty (mottygs@gmail.com)

**Abstract.** An understanding of the role of ice crystals in the cirrus cloud is significant on the the radiative budget of the planet and consequent changes in the temperature. The structure and composition of the cirrus is affected by the microphysical parameters and size and fall speed of ice crystal inside the clouds. In this study, the structure and dynamics of tropical cirrus clouds were analysed by the microphysical characterisation using the data obtained by the ground based lidar system over the tropical site Gadanki [$13.5^0$ N, $79.2^0$ E], India for a period of 6 years from 2005 to 2010. The observed clouds have optical depth within the range 0.02 to 1.8, lidar ratios are in the 20 to 32 sr range and depolarisation ratio varies between 0.05 and 0.45. The altitude and temperature dependence of the parameters, their interdependence and the fall velocity – effective size analysis were investigated. Dependence of the microphysical parameters on the ice fall velocity which is critical for climate change was also analysed. The same are compared with the CALIPSO satellite based CALIOP lidar observations.

Keywords: ground lidar; CALIPSO; cirrus optical depth; lidar ratio; depolarisation ratio; crystal shape; fall velocity.

## 1. Introduction

In the high altitude cirrus clouds, ice crystals appear in a variety of forms and shapes, depending on the formation mechanism and the various atmospheric conditions (Liou, 1986; Rogers, 1976; Heymsfield and Platt, 1984). In cirrus clouds, at temperatures lower than about $-45^0$ C, ice crystals form and exist as mainly nonspherical particles (Petrenko and Whitworth, 1999). Moreover, high altitude clouds in the altitude range between 8 to 20 km, have an important place in sustaining the energy budget (Wallace and Hobbs, 2006) of the earth atmosphere system by interacting with solar radiation (Stephens, 1990). Ice clouds reflect solar radiation effectively back to space, called albedo effect and absorb thermal emission from the ground and lower atmosphere, through the greenhouse effect (Stephens, 1990). Inspite of their occurrence hight and temperature, the microphysical conditions have fundamental implications in terms of radiative transfer (Liou and Takano, 2002). There for it is significant to analyse the microphysical properties of cirrus clouds including their structure for estimating their radiative properties accurately (Cess et al., 1990).

Earlier extensive experimental studies analysed the importance of cirrus clouds in the radiation budget using various techniques employing lidars (Liou, 1986; Prabhakara et al.; 1988; Ramanathan et al., 1989; Sassen et al., 1989; Wang et al., 1996; Hartmann et al., 2001; Stubenrauch et al., 2010; Motty et al., 2015, 2016). Heymsfield and McFarquhar (1996) found that cirrus clouds distributed in the tropics and play an important role in radiative effects. Prabhakara et al. (1993) suggested that the greenhouse effect produced by the optically



thin cirrus act as a significant factor in maintaining the warm pool. Heymsfield and Miloshevich (2003) have shown that top of the cloud layers composed of thin ice crystals, whereas the lower parts consists more thick crystals. Wang et al. (1994) and Stubenrauch et al. (2010) stated that tropical cirrus clouds have strong potential to impact the tropical and global climate. Recent research by Kulkarni et al.(2008) and Vernier et al. (2015) shows that aerosols in the tropical tropopause layer (TTL) act as ice condensation nuclei and is higher during monsoon periods.

The vertical profiles on cirrus formations over a local station can be obtained from the ground based lidar system while for global coverage observations using the Cloud Aerosol Lidar with Orthogonal Polarization (CALIOP) onboard the Cloud Aerosol Lidar and Infrared Pathfinder Satellite Observation (CALIPSO) satellite are widely used (Dessler et al., 2009; Meenu et al., 2010). During the last few years, significant efforts are being pursued to study the cirrus characteristics using the ground based lidar system over the tropical station Gadanki (Sivakumar et al., 2003,Parameswaran et al.,2003 and Krishnakumar et al.,2014), but mainly for deriving the general features and their variations in different periods of the year. Sivakumar et al. (2003) found that the various cirrus formations are closely related to the tropospheric temperature. Parameswaran et al. (2003) stated that for the cirrus covered region, the decrease in the environmental lapse rate could possibly be attributed to the cloud induced IR heating. Also according to Krishnakumar et al. (2014), a notable dependence is observed between the ice crystal morphology in the clouds and the various dynamical conditions of the prevailing atmosphere. Thus the ice composition and the microphysics of cirrus can be understood by using the lidar data on their scattering properties in deriving the quantitative values of the extinction coefficient, optical depth, depolarisation properties, lidar ratio, effective size of ice crystal and fall velocity in order to analyse the radiative effects.

The major objective of this paper is to contribute towards improving the understanding of the radiative effects of cirrus clouds in terms of their microphysical parameters over the Indian tropical station, Gadanki [$13.5^0$ N, $79.2^0$ E], using the ground based NARL lidar system and the Caliop observations on a seasonal mean basis along with their interdependence. Cirrus cloud observations on 152 days during the period 2005-2010 were analysed using the data from both the lidar techniques and the results obtained were compared with the earlier reports.

## 2 Data sources and Analysis

### 2.1 Ground based lidar (GBL)

The ground based data obtained from the lidar system installed at the National Atmospheric Research Laboratory (NARL), Gadanki ($13.5^0$ N, $79.2^0$E) for the period (2005-2010). The further specification for the ground based lidar system are the same as employed in Motty et al. (2015, 2016), and the following text is derived from there with minor modification. The NARL lidar is a monostatic, biaxial duel polarisation system which transmits Nd: YAG laser pulses of wavelength 532 nm at a rate of 20 Hz (50 Hz since 2007). Each pulse has pulse energy of 550 mJ (600 mJ since 2007) and pulse duration of 7 ns. The laser beam emerging with a divergence of 0.45 mrad from the source and to reduce the divergence to < 0.1 mrad before transmitting vertically into the atmosphere, the beam is expanded using a 10X beam expander. The backscattered photons from the atmosphere are collected by two independent telescopes. One of the telescopes is designed to collect





the backscattered photons from the air molecules above 30 km in the atmosphere (called Rayleigh receiver) and
the other is designed to collect the backscattered photons from altitude below 30 km to study aerosols and
clouds (called Mie-receiver). The Mie-receiver contains a Schmidt-Cassegrain telescope of 35.5 cm diameter
with a field of view of 1 mrad. In order to eliminate the unwanted background noise from the received signal, a
narrow band interference filter with wavelength centred at 532 nm and a full-width at half-maximum of 1.1 nm
is placed in front of a polarizing beam-splitter. The polarizing-beam splitter splits the beam into parallel and
perpendicularly polarized beams which are then detected by two identical orthogonally aligned photomultiplier
tubes. The counting system consists of a Multi-Channel Scalar card and the photon counts are accumulated in
300m resolution bins and integrated for 4 min. Lidar data were collected only during the nights that are free
from low-level clouds and rain.
**2.2 Satellite based lidar (SBL)**
The CALIOP onboard the CALIPSO satellite provides high- resolution observations of the vertical distribution
of clouds and aerosols and their optical properties along the satellite track (Winker et al., 2007). The following
description about the lidar system is obtained from Motty et al., (2015, 2016) with minimal changes. In order to
compare the properties of cirrus clouds obtained from NARL lidar, level-2, 5 km cloud layer and cloud profile
data of product version 3 obtained for a grid ($5^0$ N - $20^0$ N; $60^0$ E - $85^0$ E) during the period of June 2006–
December 2010 was used (Motty et al.,2016). CALIOP is a near-nadir viewing space-based, dual-wavelength,
dual-polarization, three channel elastic backscatter lidar that transmits linearly polarized laser pulses having an
average pulse energy of 110 mJ both at first (1064 nm) and second harmonic (532 nm) wavelengths of Nd:
YAG laser (Winker, 2003; Hunt et al., 2009; Winker et al., 2009; Pandit et al., 2015; Motty et al.,2016 ). The
backscattered signal is received by a 1m diameter telescope with parallel and perpendicularly polarized channels
at 532 nm wavelengths and one parallel channel at 1064 nm (Winker et al., 2007). The present study utilizes
version 3 of the level-2 cloud layer data product from CALIOP, where the data is gridded at 5 km horizontal
resolution. The data used in this work have a vertical and horizontal resolution of 60 m and1000 m respectively
in the altitude region between 8.2 and 20.2 km.

In order to obtain the properties more accurately, simultaneous observations using ground-based and

space-borne lidar over a tropical station with opposite viewing geometry, night-time data collected when the
CALIOP overpasses nearby the Gadanki region were only considered. Because of the Caliop's repeat cycle of
16 days, four overpasses at most can be obtained in each month, with 2 each daytime and night-time overpass.
During the observation period a total number of 152 data files were collected from the NARL lidar system (
Motty et al.,2016) and 116 data files (Caliop observations are available from 2006 June onwards) from the
nocturnal observations obtained using CALIOP in the region selected around Gadanki. In this study the Calipso
observations with cloud top altitude greater than 8 km and those with CAD score in the range of 80–100 were
only considered.

**2.3 Method of analysis**
Fernald's inversion method is used to derive the extinction coefficient for the ground based NARL lidar data in
the region up to 20 km. The methods are the same as employed in Motty et al.,(2016) . Cloud optical depth



(COD, τc) is calculated by integrating the extinction coefficient from cloud base to its top and is an important
parameter depends on the composition and thickness of the cloud. The following estimations of τ are obtained
by Sassen et al. (1992) from their visual appearance: τc ≤ 0.03 for sub visible, 0.03 < τc ≤ 0.3 for thin, τc > 0.3
for dense cirrus clouds.The depolarisation ratio, δ(r) within the cloud indicates the phase of the cloud and
thereby to identify the type of ice crystals present within the cloud. Most of the tropical cirrus clouds are
composed of non-spherical ice crystals and will cause significant depolarisation.The lidar ratio (sc) depends on
the structure and properties of ice crystals within the cirrus and as such it is range dependent. The range
dependent lidar ratio is obtained using the method suggested by Satyanaryana et.al. (2010).
Also the Caliop parameters of cloud layers are obtained directly from CALIOP 5 km cloud layer data products.
The extinction coefficient as well as the IWC depends upon the particle size distribution of the clouds. The IWC
can be determined from the extinction measurements by using the relation provided by Heymsfield et al. (2005).
The effective diameter provides additional information for the ice cloud radiative characteristics and can be
obtained from the equation by Heymsfield et al. (2014) as a function of IWC, σ and the density of solid ice (ρ).
In order to analyse the fall velocity of ice crystals in terms of effective size, an aerodynamic equation
suggested by Mitchell et al., (1996) was employed.
**4 Results and discussion**
**4.1. Microphysical statistics of tropical cirrus**
The extinction coefficient, optical depth, lidar ratio and depolarization ratios are considered to be of special
importance since they are related to microphysics of the ice crystals contained on cirrus clouds. The
microphysical properties as well as the mid cloud altitude and temperature of the cirrus clouds together play an
important role in the radiative properties (Sunil kumar et al., 2008; Seifert et al., 2007). Here the climatology of
the cirrus over the tropical station Gadanki covering 152 days of lidar observations from 2005 to 2010 were
analysed. The observation period is divided into the prominent seasons of the Indian subcontinent namely winter
(December,January,February), summer(March,April,May),South West monsoon(June,July,August) North East
monsoon(September,October,December).
**4.1.1. Extinction coefficient**
The extinction coefficient (σ) provides the information on the influence of scatters on the radiation. Figure1
shows the contour plot of seasonal variation of σ derived from ground based as well as satellite based lidar
observations.
It can be seen that there is significant variation in the values of σ during the period of observations for ground
based system. It is found that the σ value ranges between $5.0 \times 10^{-3}$ to $8.91 \times 10^{-6}$ m$^{-1}$ and it is noted to be
highest during summer season having the range of value $1.5 \times 10^{-4}$ to $1.7 \times 10^{-6}$ m$^{-1}$. During winter the σ value is
in the range of $2.0 \times 10^{-6}$ to $2.5 \times 10^{-5}$ m$^{-1}$ range. The σ value during South-West Monsoon period is found to
be higher than North East Monsoon period having σ value ranges between $1.5 \times 10^{-6}$ to$1.7 \times 10^{-4}$m$^{-1}$and $1.0$
$\times 10^{-6}$ to $1.5 \times 10^{-5}$ m$^{-1}$ respectively. The satellite based Caliop lidar results also show similar behaviour and





the σ values ranges between $1.7 \times 10^{-4}$ to $7.8 \times 10^{-6}$ m⁻¹. During the summer period σ ranges from $5.8 \times 10^{-4}$ to
$4.6 \times 10^{-3}$ m⁻¹ and during winter from to $4.5 \times 10^{-4}$ to $3.6 \times 10^{-3}$ m⁻¹. Similar to ground based observation during
south west monsoon, Caliop observation also shows higher σ during south west monsoon period than the north
east monsoon periods having values ranges from $6.2 \times 10^{-4}$ to $5 \times 10^{-3}$ m⁻¹ and $3.9 \times 10^{-4}$ to $3.1 \times 10^{-3}$ m⁻¹
respectively.
Figure 2(a & b) shows the trend of the variation of average extinction coefficient with mid cloud height and
temperature from both observation techniques. In both cases the observed σ value does not show any clear
altitude dependence, but from the figure (2a), it is clear that most of the clouds formations in the altitude range
13-15 km range from both observation techniques. The figure (2b) shows the temperature dependence of σ
values and the points represent the average value of σ with mid cloud temperature and it can be seen that
σ decreases with the temperature and the most favourable cirrus occurring temperature ranges between -70 to -
55 $^0$C.

### 4.1.2. Optical Depth

The statistics of seasonal variation of optical depth by the two measurement techniques are shown in
figure 3.
The contour plot of seasonal variation of cirrus optical depths distribution for each season is depicted in
the figure 3. From fig. 3(a), 59% of the observed clouds were optically thin clouds with $\tau < 0.3$, 33% were sub-
visual clouds with $\tau_c < 0.03$ and only 8% were thick clouds with $\tau > 0.3$. Also in the region of 14 km and above,
where the cloud frequency is high, it is noted that 50% of clouds were sub-visual ($\tau_c < 0.03$) and 36% were
optically thin ($\tau_c < 0.3$) and the remaining are thick clouds. During the summer season τ ranges within 0.005 to
1.4 and during winter from 0.005 to 1.   During the south west monsoon period, most of the observed τ values
are for thin clouds ranges between 0.01 and 1, and thick clouds were observed in all other seasons. North east
monsoon clouds give the τ within 0.002 to 0.87. The uncertainty related to cloud optical depth estimation from
CALIPSO data is less than 50% (Winker et al., 2007). The CALIPSO observations show that these clouds have
an average optical depth 0.15-1.7km. Figure 3(b) shows that from the satellite observation, 44% of the observed
clouds were optically thin clouds and the rest were thick clouds. The 80% of the observed clouds during the NE
monsoon period were optically thin. The seasonal optical depth distribution behaviour shows the same pattern in
the range 0-1km distributed over the observation periods for the two measurement techniques. The monsoon
periods have clouds with high optical depth in ground based measurement but the values are less when
compared to CALIPSO measurement. This may be due to the higher convective activities at tropics during the
monsoon periods.

The scatter plot of variation of optical depth with mid cloud height and temperature are shown in Fig.
4( a & b). From the height dependence of optical depth (fig. 4(a)), it can be inferred that optical depth increased
with the cloud height in the 11-15-km range and then decreases. Generally, the cirrus clouds that exist in the
tropopause region are optically thin (Nee et al., 1998). Figure 4(b) shows the temperature dependence of optical
depth and from that most of the thin clouds are having low temperature (below -68$^0$C) and cloud formations are
widely observed in the range -70$^0$C to -55$^0$C.  As the temperature increases the optical depth also increases. The
clouds observed in the tropical tropopause region are having an average temperature of $\approx 75^0$ C, are normally



optically thin (Nee et al., 1998). Also it demonstrates a slightly positive dependence with the mid cloud
temperature which disagree with Wang et al. (2013) but in agreement with results of *Sassen* et al. (1992) which
shows a positive dependence with the mid cloud temperature.

### 4.1.3. Lidar Ratio

The correct values of extinction to back scatter ratio, commonly known as lidar ratio depicts the idea of
ice crystals in the cirrus clouds and reflect the cloud characteristics in the corresponding height region (Das et
al., 2009; Chen et al., 2002). According to Heymsfield and Platt (1984), in cirrus clouds at temperatures $\leq$-50 $^0$C
and altitude 12 km contains different kinds of ice crystals.
Figure 5 shows the contour plot of seasonal distribution of lidar ratio during the period of observation.
Above 12 km, the lidar ratio values are mainly distributed between 20 to 30 sr ranges. During monsoon periods,
LR shows relatively higher values which varies between 20-27sr. By the satellite based observation, the LR
values varies between 24-32 sr and during NE monsoon periods, all the observed values are in the range 25-26
sr. The calculated lidar ratio can be compared with the previously reported results.  Grund and Eloranta (1990)
reported the LR values about15–50 sr using high spectral resolution lidar during the FIRE field .Sassen and
Comstock (2001) calculated the mean mid-latitude LR as ~24±38 sr with a median value of ~27 sr using
LIRAD. Pace et al.  (2002)  found the mean LR as ~19±33 sr for the equatorial cirrus. Also Seifert et al. (2007)
derived the mean LR during monsoon over tropics and derived a mean value of 32±10 sr. Giannakaki et
al.(2007) derived LR  as 28±17 sr for  mid-latitude cirrus . Das et al.(2009) determined the mean LR value of
23±16 sr by using the simulation of lidar backscatter signals. Statistically; the above results are in agreement
with the present study.
Figure 6 (a) shows that the lidar ratio varies with mid cloud height randomly. It was found that within 12.5-15
km, the lidar ratio values are distributed mainly between 20- 28. The lower and higher clouds shows relatively
smaller LR values and over 13.5-14.5 range, higher LR values of the range 28-32 sr were observed. This
indicates that most of the observed clouds consist of hexagonal type crystals (Sassen et al., 1992) for both
observations. As in Fig. 6(b), lidar ratio appears to vary with mid cloud temperature with no clear tendency.
This can be due to the variations in the ice crystal properties at the particular temperature range or may be due to
the heterogeneous cloud formations over the region (Das et al.,2009). But in some cases, high LR values are
observed at lower temperature region which indicates the presence of thin cirrus clouds.

### 4.1.4. Depolarisation Ratio


Depolarisation ratios are influenced by the inhomogeneity of ice paricles in cirrus clouds in the lidar
analysis. The depolarisation values for the water droplets are smaller rather than the ice particles.  Figure 7
shows the seasonal distribution of depolarisation ratio and their height dependence and the wide range of
variation observed can be due to the heterogeneous nucleation process of cirrus, which results in different cloud
composition or may be due to the aspect ratio changes (Das et al., 2009). For ground based measurements, the
depolarisation ratio ranges within 0.04-0.45 as in fig.7(a) and 0.3-0.5 for satellite based measurement techniques
as in fig.7(b) and in both cases, maximum values are shown  during summer period. Between 11-16km range,



the higher depolarisation values are observed (Motty et al., 2016) and above 16 km, smaller values < 0.2 are
observed may be due to the presence of super cooled or mixed phase of water (Sassen, 1995) and the
horizontally oriented ice crystals increases the same (Platt et al., 1978). The NARL Lidar observation showed
that during winter periods depolarisation varies between 0.05-0.3 and for monsoon periods it is 0.04-0.3. The
highest depolarisation values are observed during the summer periods. The depolarisation variation by the
satellite observation varies in the range 0.3-0.5 and for summer all observed values were higher and show
relatively lower values during NE monsoon periods. During monsoon, since the cloud condensation nuclei are
relatively higher the water content is more in the clouds.
Figure 8(a) presents the scatter plot of the dependence of average value of depolarization ratio on the mid cloud
height and the values are found to be scattered for both the measurement techniques; but an increasing tendency
is observed between 10-16 km. In most of the cases, depolarization ratio values ranges within 0.05 to 0.45.
Figure 8(b) shows depolarisation ratio plotted as a function of mid cloud temperature and  most of the ice clouds
having depolarisation values above 0.3 show an increasing tendency with decreasing temperature within −80 to
−30 ∘C (Motty et al.,2016). For water clouds, depolarisation values ranges between 0-0.09 and it doesn't show
any temperature dependence. Since at higher altitude region, the cooler cirrus clouds are having bigger sized ice
crystals results higher depolarisation values and are in agreement with by Chen et al. (2002).
**4.2. Interdependence of the optical properties**
The dependence of microphysical characteristics of cirrus clouds over Gadanki were further investigated.
Figure 9(a) depicts the scatter plot of optical depth variation with lidar ratio. The lidar ratio of thin cirrus was
observed in 14-28 sr range and has the maximum value obtained at an optical depth range between 0.05 and 0.3,
which agrees with Wang et al. (2013). As the optical depth increases (thick clouds), the lidar ratio decreases and
is more clear in the satellite based observation. Also most of the optically thin cirrus shows lower lidar ratio
values. The variation of depolarisation ratio with optical depth is shown in figure 9(b) and from that clouds
having large depolarisation values are observed with smaller optical depth. From the figure, it is clear that the
sub-visual clouds show higher depolarisation values. The points with lower optical depth and higher
depolarisation values indicate the presence of thin cirrus clouds. In both the observation techniques, the points
denoting higher optical depth and higher depolarisation ratios show the occurrence of thick cirrus clouds. The
points with smaller depolarisation values are for water clouds. Some of the previous studies revealed a relation
between the particle depolarization ratio and particle lidar ratio, but found differing interdependencies.
According to Reichard et al. (2002), the particle depolarization ratio splits into two branches and with increasing
particle lidar ratio values, the difference of these two branches decreases. According to Chen et al. (2002), the
particle depolarization ratio splits into two branches, but with increasing particle lidar ratio values, the
difference of these two branches increases. Later by Sakai et al. (2006), the values for particle depolarization
and particle lidar ratio are enclosed between two vertices and are possible to distinguish the phase and
orientation of randomly oriented ice crystals by measuring the particle depolarization and particle lidar ratio.
Figure9(c) shows the relation between depolarisation ratio and lidar ratio. For the present study, it does not show
any particular dependence.
According to the hexagonal crystal classification by Sassen et al. (1992), the observed data can be classified as
in fig.10. Here for the ground based observation, 51.20% of the cloud containing hexagonal thin plate ice



crystals, 30% were hexagonal thick plate crystals and 18.8% were long column type. From the satellite
observation, 52.8% of the cloud crystals were long column, 42.1% were thin plates and only 5.2% were thick
plate type crystals. The probable size and shape distribution of ice crystals in the cirrus clouds can be further
analysed clearly by calculating the effective diameter and fall velocity analysis.
**4.3 Effective diameter and fall velocity analysis**

Das et al. (2010) obtained the possible ice crystal formations with respect to the fall velocity – effective

diameter relation derived by Mitchell (1996). Here the average value of effective diameter is obtained using the
equation by Heymsfield et al. (2014) as a function of IWC, σ and the density of solid ice (ρ) and according to
the diameter obtained, the possible shapes are selected as suggested by Mitchell (1996). The obtained effective
diameters were estimated by averaging the diameter obtained during each season for the observed year. In the
present observation, ice crystals of various shapes, mainly hexagonal plates (HP) ($15 \leq D \leq 100$ and $100 \leq D$
$\leq 3000$) and hexagonal columns (HC) ($100 \leq D \leq 300$ and $D > 300$) have been analyzed and the fall velocity is
obtained using their corresponding equations obtained by Das et al.(2010). These obtained ice crystal shapes are
similar to the previous classification as by Sassen et al. (1992) and are commonly found in cirrus clouds. The
seasonal average of the obtained parameters by the two measurement techniques are summarized as in Table
1and 2.
Several earlier studies reported the size of most of the cirrus ice crystal to be about several hundred microns and
falls with velocity between 30 and 100 cm/sec (Mitchell, 1996; Heymsfield, 2003; Deng and Mace, 2008; and
Das, 2010). The results of the present study are in agreement with the previously obtained values.
The effective size of ice crystals in cirrus usually increases with mid cloud temperature as in figure 11 indicates
the presence of smaller sized ice crystals at lower temperature region. The lower temperature has been suggested
for homogeneous ice nucleation (Kärcher and Lohmann, 2002). The results obtained are similar for both the
ground based and satellites based observation techniques and are in agreement with Das et al. (2010),
Heymsfield et al. (2000), and Chen et al. (1997).

The fall velocity, the rate at which an ice crystal falls through a cloud, is dependent on its mass, size and shape
and thus can be used for effective size analysis. From the scatter plot of fall velocity variation with the mid
cloud temperature as in figure12, the fall velocity is found to be broadly distributed and show an increasing
tendency with temperature. This is in agreement with the results obtained by Das et al. (2010) observed over
Chung-Li using the lidar measurements. As depicted in figure 12(b), the fall velocity varies linearly with the
effective size of cirrus crystal. Figure 12(c) shows the dependence of fall velocity on the cirrus optical depth
measurements. For larger ice crystals, the chances for homogeneous nucleation are lesser and the sedimentation
rate increases. The smaller sized crystals having low fall velocities undergo homogeneous nucleation processes
and thus the cloud remains for longer in the upper atmosphere. Also the water vapour in that region is higher
and it acts as a blanket and prevents long-wave radiation being emitted by the Earth from escaping into space,
enhancing warming. The effective size distribution of ice particles which decides the lifetime of the cirrus
clouds has strong impacts on the cloud radiative forcing due to its influence on cirrus cloud coverage. Here, the
fall velocity of cirrus ice crystals observed are relatively high, which may indicate that the ice particles will fall





out rapidly. Therefore, it can be said that the lifetime of the cirrus clouds in the tropics will dissipate faster and
causes LW dominance and thus have a significant effect on the radiation budget.
**Conclusion**
The important microphysical properties such as extinction coefficient, optical depth, lidar ratio and
depolarisation ratio for cirrus clouds obtained during 2005-2010 were analysed using the observations made by
the ground based lidar system at NARL Gadanki ($13.5^0$ N, $79.2^0$ E) and are compared with the available night
time observations from the CALIOP on board the CALIPSO satellite. The dependence between these quantities
and its relationship with mid cloud height and mid cloud temperature were also obtained. The radiative effects
of obtained cirrus clouds were discussed by the effective size – fall velocity analysis of the cirrus ice particles.
The inter comparison of measurements by the two techniques showed that the satellite measurements match
very closely with the ground based data. However, small differences were noticed since the observations were
not being exactly obtained at the same place and the sampling frequencies were also different. Some of the
generally observed results from both observation techniques are listed which are as follows:

• The cirrus extinction coefficient ranges from $5.0 \times 10^{-3}$ to $8.91 \times 10^{-6}$ $m^{-1}$ and $1.7 \times 10^{-4}$ to $7.8 \times 10^{-6}$
$m^{-1}$ for ground based and satellite based measurements respectively and it decreases with the
temperature.
• Cirrus optical depth varies from 0.02 to 1.8 ranges with most frequent occurrence being in the range
0.04 to 0.3. It was noted that optical depth increases with temperature of the mid cloud within the range
$-50^0$C to $-70^0$C and decrease with temperature outside this range. Within 11-15km range, the optical
depth increases with height and decreases for higher altitudes.
• The lidar ratio varies with mid cloud height randomly, and within 10 to 20 km the lidar ratio values are
distributed in the range 20 to 32 sr. The lower and higher clouds show relatively smaller LR values and
within 13.5 to 14.5 range higher LR values of the range 28 to 32 sr were observed. Lidar ratio varies
with the mid cloud temperature with no clear trend.
• Depolarization ratio was observed to increase from 0.05 to 0.45 with height between 10 and 16 km and
tend to decrease with the temperature varying from $-80$ to $-30$ $^0$C.
• Most of the optically thin cirrus shows lower lidar ratio values and are scattered among 14-28 sr range.
As the optical depth increase (thick clouds), the lidar ratio decreases.
• The clouds having large depolarisation values are with lower optical depth. Thin cirrus have lower
optical depth ($\tau \leq 0.3$) and higher depolarisation values (0.3 to 0.35) whereas the higher optical depth
($\tau > 0.3$) and higher depolarisation ratios (0.35 to 0.45) shows the occurrence of thick cirrus clouds and
also the smaller depolarisation values (below 0.1) are for water clouds.
• The presented results are used to understand possible hexagonal crystal formations inside clouds using
the depolarisation ratio- lidar ratio analysis. The ground based measurements gives the occurrence of
more hexagonal thin plate crystals (51.20 %) during the observation whereas the satellite based
measurements shows the presence of higher long column crystal formation (52.80 %).



- • The effective size of the ice crystal is lower at colder temperature and thus showing the large size at the
- warmer temperature.
- • The fall-velocity of the ice crystals increases with temperature, indicating the influence of particle
- growth in cirrus coverage. The higher values of ice crystal fall velocity in cirrus clouds observed
- indicate rapid fall out of the ice particles which causes the faster dissipation of cirrus clouds over
- tropics causes the LW dominance and thus have a significant effect on the radiation budget.
- • The detailed study on the influence of the possible hexagonal crystal formation on the characterisation
- of radiative properties can be pursued in future.
- • The results obtained on various microphysical properties of cirrus and their interrelationship along with
- a comparative study of GBL and SBL data are expected to be useful in the radiative budget analysis.

**Acknowledgments**
The authors thank the National Atmospheric Research Laboratory, Department of Space, Government of
India, Gadanki, Tirupati, India for providing lidar data and also acknowledge the assistance given by NASA
Langley Atmospheric Science Data Centre for providing CALIPSO data.

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

**Table 1. Seasonal average value of properties of cirrus clouds derived with ground based lidar**



| Mid cloud height(km) | Mid cloud temperature($^0$C) | Optical depth | Depolarisation ratio | Lidar ratio | Effective particle size(µm) | Fall velocity (cm/sec) |
|---|---|---|---|---|---|---|
| 17.2 | -79.9 | 0.0173 | 0.1696 | 24.21 | 281.3 | 34.5 |
| 11.84 | -49.96 | 0.55 | 0.1568 | 20.64 | 272.7 | 33.4 |
| 13.99 | -70.75 | 0.05 | 0.1054 | 15.01 | 261.8 | 32.05 |
| 14.71 | -70.33 | 0.15 | 0.2168 | 21.84 | 265.9 | 32.5 |
| 16 | -79 | 0.028 | 0.3113 | 12.96 | 272.4 | 33.38 |
| 14.06 | -67.2 | 0.25 | 0.0521 | 20.64 | 277.1 | 33.9 |
| 11.62 | -41.51 | 0.05 | 0.0643 | 17.09 | 276.4 | 33.3 |
| 13.24 | -63.42 | 0.106 | 0.0507 | 23.22 | 398.5 | 49.5 |
| 14.48 | -66.31 | 0.01393 | 0.1234 | 17.63 | 259.6 | 31.7 |
| 14.38 | -63.5 | 0.012 | 0.0995 | 27.16 | 335.9 | 41.4 |
| 14.66 | -68 | 0.6 | 0.415 | 25.11 | 244.8 | 29.9 |
| 12.93 | -64.3 | 0.25 | 0.189 | 23.22 | 286.6 | 35.2 |
| 14.17 | -61.6 | 0.255 | 0.0408 | 26.08 | 312.9 | 38.5 |
| 14.39 | -70.16 | 0.038 | 0.0589 | 22.94 | 304.6 | 37.4 |
| 14.64 | -67.58 | 0.0106 | 0.3807 | 16.63 | 378.4 | 46.8 |
| 13.86 | -64.6 | 0.0118 | 0.1026 | 19.98 | 264.1 | 32.3 |
| 12.4 | -46.34 | 0.3 | 0.0418 | 25.73 | 360.02 | 44.6 |
| 15.1 | -62.3 | 0.2 | 0.3239 | 22.94 | 262.8 | 32.18 |
| 13.05 | -57.15 | 0.265 | 0.1505 | 20.08 | 316.6 | 39.01 |
| 13.77 | -65.66 | 0.014 | 0.1182 | 21.23 | 377.9 | 46.79 |
| 12.39 | -62.5 | 0.013 | 0.1176 | 21.9 | 288.4 | 35.42 |
| 14.76 | -67 | 0.227 | 0.131 | 22.55 | 263.1 | 32.29 |
| 14.82 | -63.75 | 0.01194 | 0.0738 | 25.62 | 274.6 | 33.7 |
| 14.1 | -54.5 | 0.75 | 0.1995 | 21.41 | 257.5 | 31.51 |






**Table 2. Seasonal average value of properties of cirrus clouds derived with satellite based lidar**





| Mid cloud height(km) | Mid cloud temperature($^0$C) | Optical depth | Depolarisation ratio | Lidar ratio | Effective particle size(μm) | Fall velocity (cm/sec) |
|---|---|---|---|---|---|---|
| 13.56991 | -61.96707 | 0.44233 | 0.44072 | 31.22479 | 338.37518 | 44.74023 |
| 13.94052 | -65.62776 | 0.23966 | 0.47192 | 29.45267 | 334.79426 | 44.26388 |
| 13.43109 | -61.88633 | 0.60375 | 0.4447 | 26.12036 | 326.65703 | 43.1413 |
| 13.91251 | -64.72676 | 0.97909 | 0.44007 | 26.19174 | 337.61668 | 44.63188 |
| 12.68714 | -55.72516 | 0.37189 | 0.39678 | 28.35552 | 371.51118 | 49.26096 |
| 14.78327 | -68.16807 | 0.28282 | 0.44113 | 25.49286 | 335.54569 | 44.35372 |
| 14.0175 | -64.31054 | 0.79158 | 0.45619 | 24.98762 | 361.22532 | 47.85666 |
| 15.59179 | -75.7787 | 0.06528 | 0.44678 | 25.09948 | 360.13977 | 47.70968 |
| 14.04506 | -63.33783 | 0.57707 | 0.43175 | 26.85669 | 334.43792 | 44.19976 |
| 13.90262 | -61.09068 | 0.61027 | 0.43411 | 25.96867 | 357.97366 | 47.40541 |
| 14.11647 | -64.21318 | 0.93512 | 0.43742 | 25.21233 | 375.99098 | 49.8676 |
| 13.37893 | -60.43045 | 0.89786 | 0.44775 | 25.94735 | 379.87508 | 50.39671 |
| 13.52856 | -58.89109 | 0.73884 | 0.40346 | 27.38521 | 363.94335 | 48.22065 |
| 13.83301 | -60.21539 | 1.0968 | 0.42088 | 26.24744 | 364.1064 | 48.24252 |
| 13.8945 | -64.61901 | 0.55583 | 0.42907 | 26.75646 | 352.39667 | 46.65443 |
| 13.06972 | -57.32295 | 0.64174 | 0.40611 | 27.57685 | 332.45256 | 43.93023 |
| 13.23237 | -56.84902 | 0.38632 | 0.3553 | 25.29714 | 311.68792 | 41.10485 |
| 12.63906 | -54.21735 | 0.59498 | 0.37918 | 27.04626 | 354.66284 | 46.96083 |














**Figure 1: Contour plot of seasonal variation of Extinction coefficient for (a) ground based lidar (GBL) and (b) satellite based lidar (SBL) observations.**







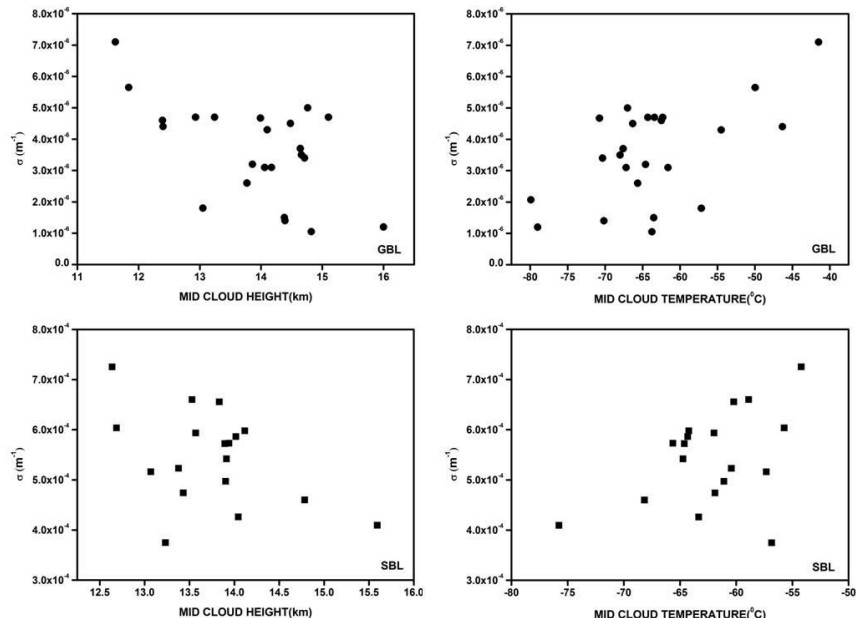


**Figure 2: Variation of extinction coefficient according to mid cloud height and mid cloud temperature based on GBL and SBL measurement techniques.**

























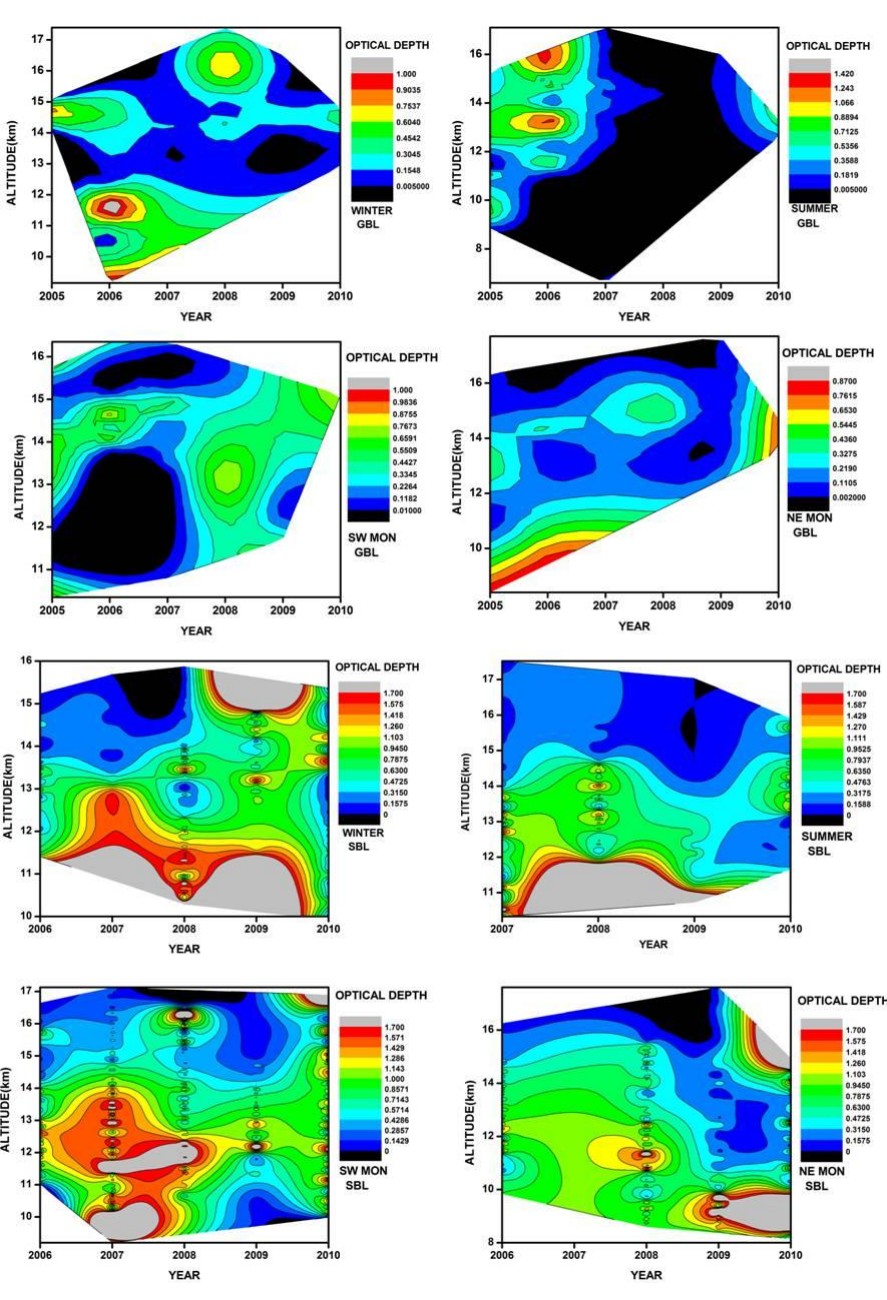


**Figure 3: Contour plot of seasonal variation of optical depth based on GBL and SBL observations.**







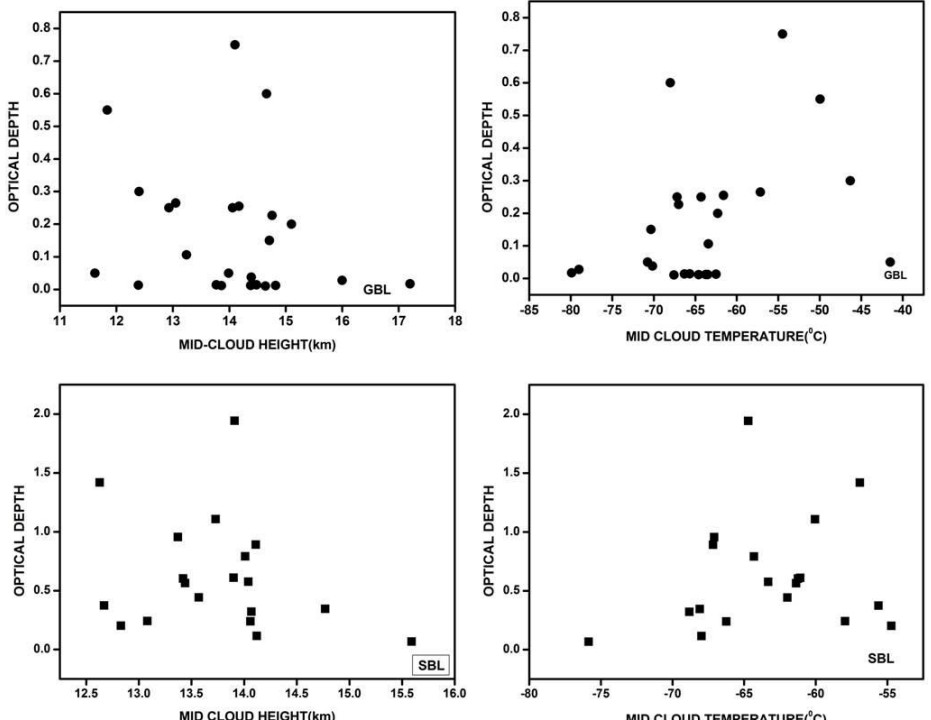

**Figure 4: Variation of optical depth according to mid cloud height and mid cloud temperature based on GBL and**
**SBL measurement techniques.**




**Figure 5: Countor plot of seasonal variation of Lidar ratio using GBL and SBL observations.**





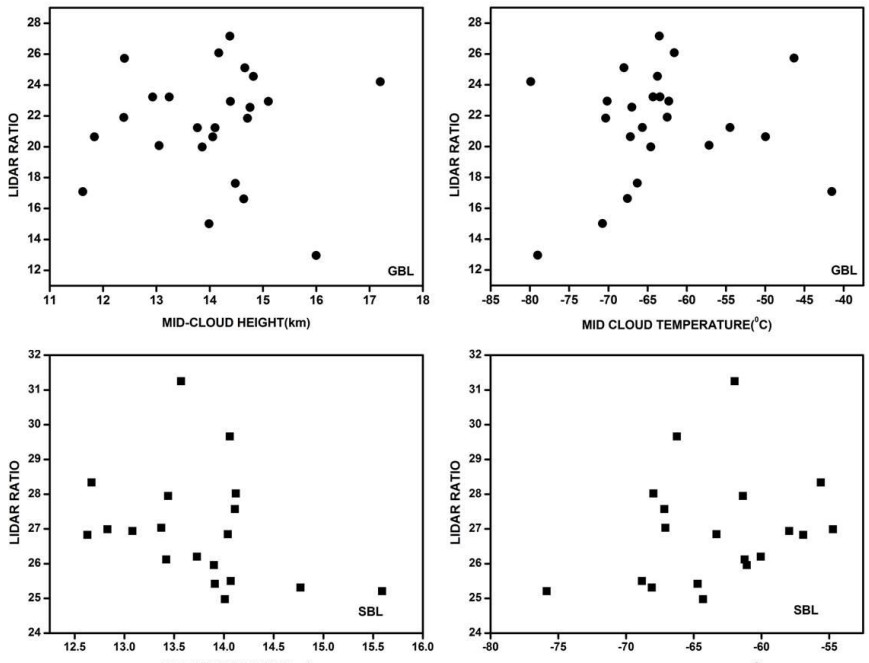


**Figure 6: Variation of Lidar ratio according to mid cloud height and mid cloud temperature based on GBL and SBL measurement techniques.**


















**Figure7: Countor plot of seasonal variation of Depolarisation ratio using GBL and SBL observations.**



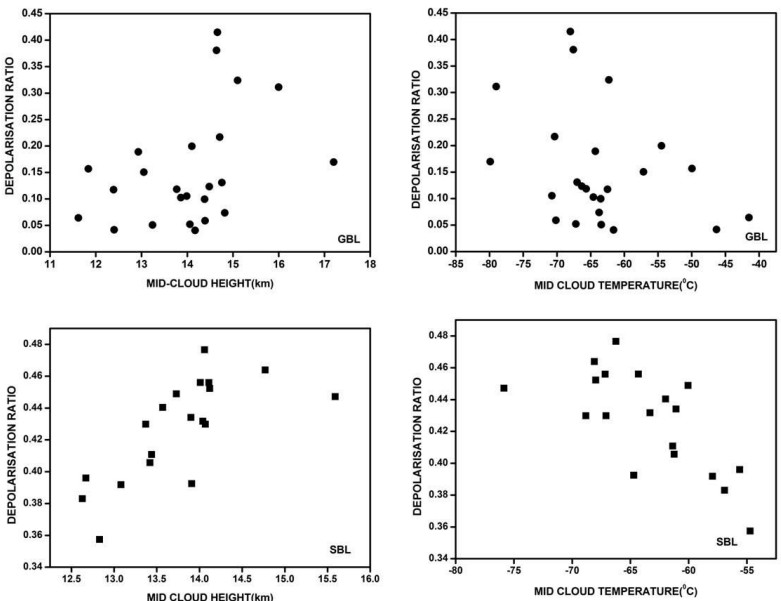

**Figure 8: Variation of Depolarisation ratio according to mid cloud height and mid cloud temperature based on GBL**
**and SBL measurement techniques.**




















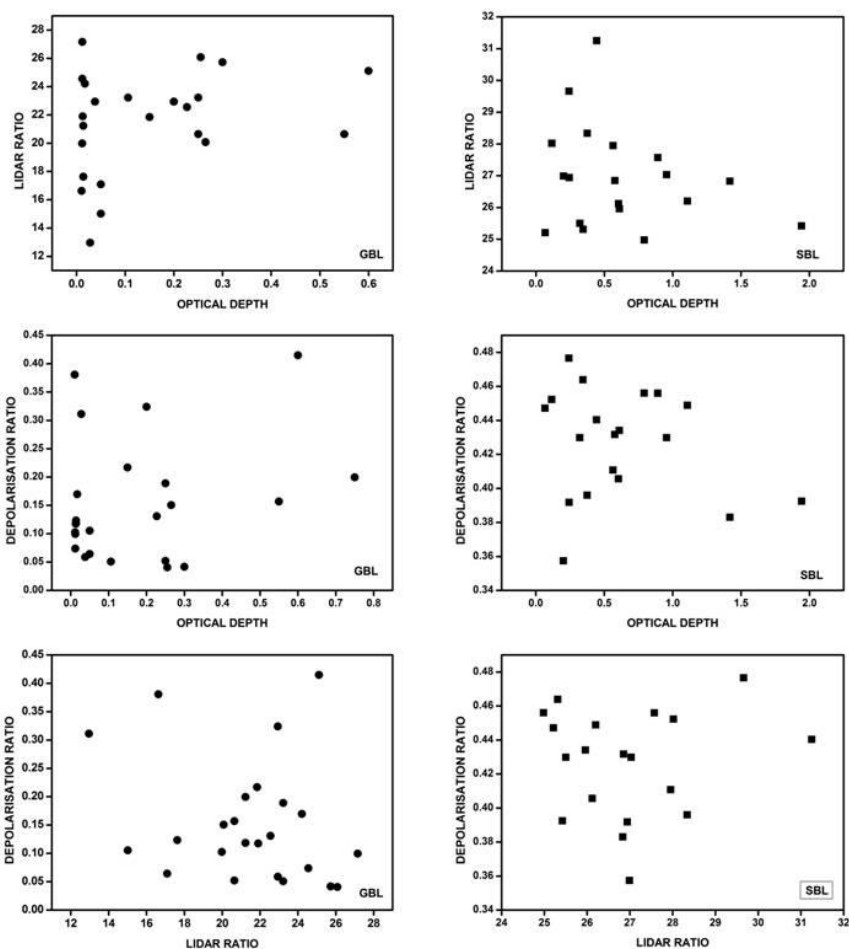


**Figure 9: The Inter dependence of (a) Optical depth, (b) Lidar ratio and (c) Depolarisation ratio based on GBL and SBL measurement techniques.**











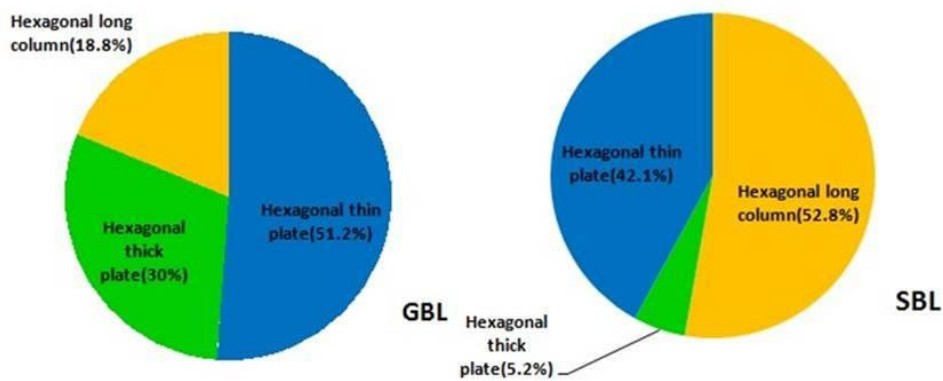


**Figure 10: Percent contribution of different types of Hexagonal crystals in the cirrus observations based on the LR and DPR values by using Ground based lidar(GBL) and Satellite based lidar (SBL).**
























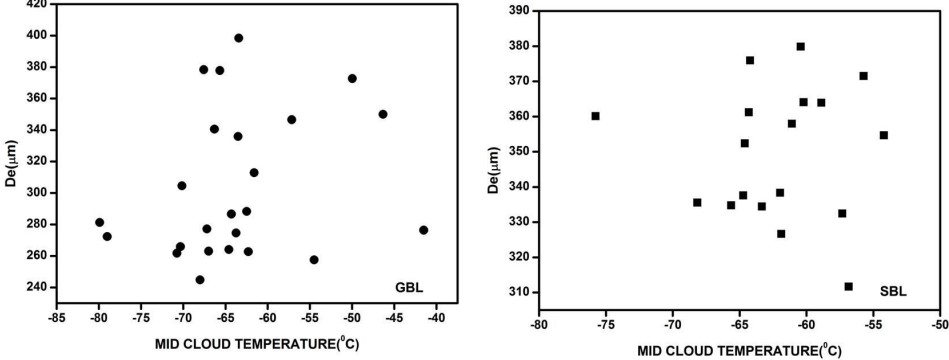


**Figure 11: The scatter plot of dependence of effective size of ice particles in cirrus clouds with mid cloud temperature by using GBL and SBL.**



























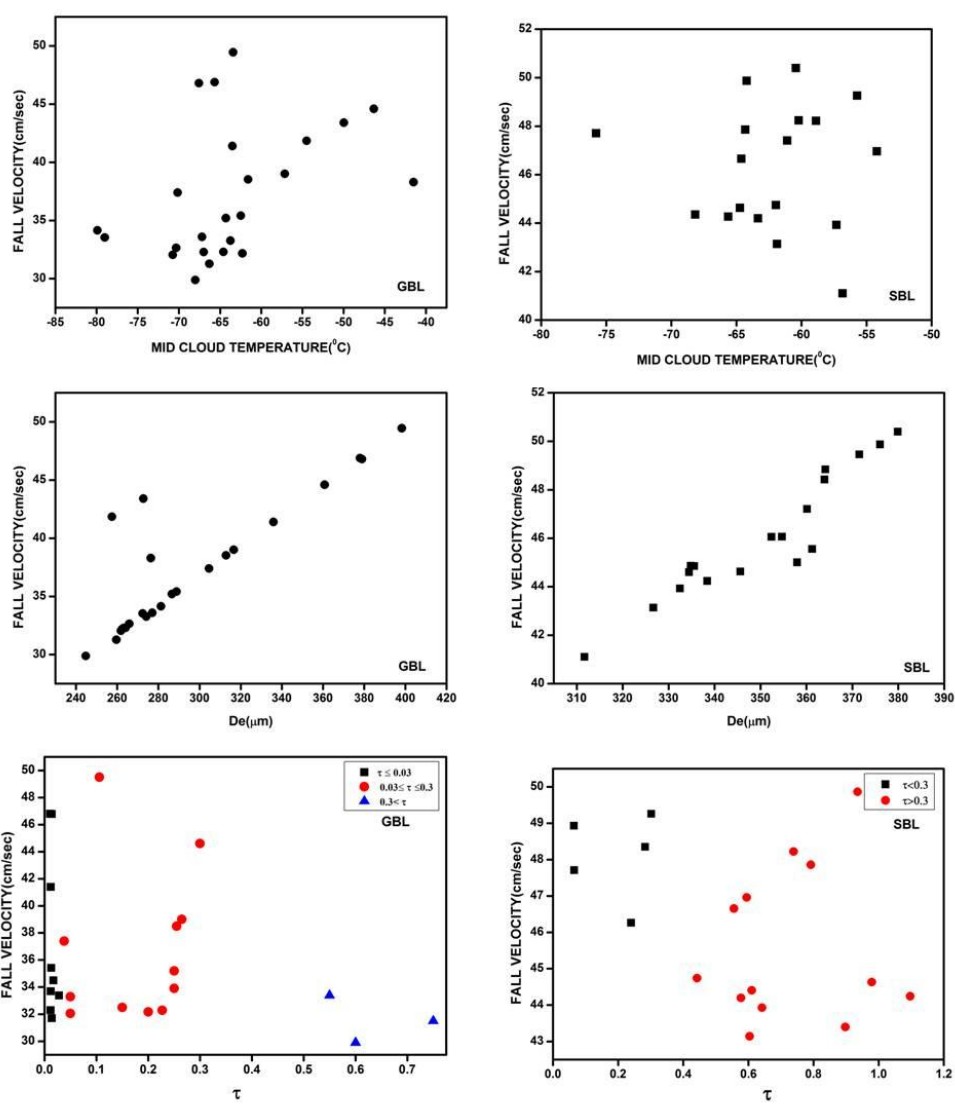


Figure 12: Dependence of fall velocity of cirrus cloud particles with mid cloud temperature , effective size and optical depth by using GBL and SBL.




