# Peer review of "Lidar studies on microphysical influences on the structure and lifetime of tropical cirrus clouds"

_Atmospheric Chemistry and Physics, 2016_

## Referee Comment (RC1) · Anonymous Referee #1 · 25 Jul 2016

The manuscript presents seasonal mean values of lidar optical properties of clouds measured at Gadanki, India over 6 years, and from the CALIPSO space lidar in a similar region, plus some discussion of inferred particle size and crystal habit. With regret, I recommend rejection of this manuscript. I encourage the authors to put additional effort into the analysis and reworking and resubmitting the manuscript. However, given the amount of additional analysis that is required, it does not seem possible to do it within just a few weeks.

Some specific comments: Here I do not comment on every aspect of the manuscript that is unsatisfactory. If the authors consider the points offered, they will be in a better position to perform a useful analysis and write an acceptable manuscript. However,

just fixing a few points in the current manuscript won't be enough.

Objectives:

First, the title and objectives should be better matched to the actual content of the manuscript. The introduction states that the objective of the paper is to contribute to the understanding of the radiative effects of cirrus clouds in terms of their microphysical parameters over Gadanki. Actually, what's presented is a catalog of optical properties (extinction, optical depth, lidar ratio) measured by lidar, not microphysical properties (except for effective diameter which I'll come to below). There is no information about radiative effects or about the "structure and lifetime" (from the title), except a few sentences of pure speculation at the very end.

Methodology:

The methodology section is very incomplete. More than just references are required. Please show equations and discuss how to apply them. How is the fall velocity calculated? (show examples?) Show equations for how the effective diameter is calculated. Describe the methodology for inferring the ice crystal shape from the lidar ratio measurements - this references Sassen et al. 1992, but in that paper I can find no prescription for making this inference, only theoretical calculations for a few single shapes. For all the quantities that are inferred or calculated rather than measured from the lidar (IWC, effective diameter, crystal shape), what else needs to be assumed? What are the uncertainties/systematic errors for these calculations?

Where does the temperature measurement come from? Are there other ancillary datasets?

Also, there is confusing and incomplete descriptions of the CALIOP dataset in use. Most important, there is no discussion of how the CALIOP lidar ratio is calculated. In most cases, CALIOP cannot measure the lidar ratio and infers a value based on climatology. In some cases, however, if there is an isolated cloud with clear air both

above and below it, it is possible to calculate a lidar ratio directly. Is the current analysis limited to clouds that have a calculated (rather than inferred) lidar ratio? It would not make sense to draw any scientific conclusions from the CALIOP lidar ratio if the inferred lidar ratios are used in the analysis.

For the ground-based dataset, how were the cloud data separated from non-cloud data? (Is it possible they were NOT separated and that is why there are so many low seasonal mean depolarization ratios?)

Analysis:

It is more difficult than necessary to make any comparisons from the figures since all the ranges on all subplots tend to be different. Please make it easier on yourself and your readers and equalize the ranges for different panels within a figure. The contour plots are almost completely useless because of this. The contour plot color scales are even more difficult to compare with each other because they are neither linear nor logarithmic. Better to use one of these two standards.

I'm also very confused by the correspondence between the line plots and the scatter plots. For example, the satellite (SBL) portion of Figure 6 shows no lidar ratios outside the range 24-32 sr. However, the SBL portion of Figure 5 shows a minimum value of something below 20.38 sr and a maximum of something greater than 73.25 sr (both in SW Mon). Why do these values not appear in Figure 6? The text says "during the NE monsoon period, all the observed values are in the range 25-26 sr" but there is at least a little bit of green in Figure 5 (NE Mon SBL), which in that panel corresponds to values greater than 31.03 sr.

What are the different rows in table 1 and table 2? The table caption says these are seasonal means (not individual clouds) so are the rows for different seasons and years? They should be labeled.

From the beginning, seasonal means are presented. It is not clearly stated but seems

likely that all the contour and scatterplot figures 1-12 present seasonal means (for individual years) rather than data for individual clouds. So, even though the text describes these as graphs of seasonal variation, they are actually graphs of interannual variation showing only seasonal means. This seems like a mistake, given the questions you are trying to answer. For example, on page 5, line 167, you describe figure 3(a) as saying 59% of the clouds were optically thin. This is not accurate. Rather, 59% of the means correspond to optical depths of thin clouds, a conclusion that might be interesting but is different from what's stated. Later, line 170-174 you give ranges of optical depth values for "clouds" but information for individual clouds was lost in the process of taking seasonal means. In reality, these ranges are only the ranges for the means, not the ranges for clouds. In the text, this could be considered just sloppy language, but the same objection holds for all the parts of your analysis where you are looking for correlations. To use seasonal means, you should first make sure that they are representative of the behaviors you want to describe. It seems that the variability you want to explain is lost in the process of collapsing all your data down into a few seasonal means. Too much information about cloud variability was lost to still have the ability to determine the dependence of the cloud height or cloud optical depth on temperature (Figure 2) or of lidar ratio or depolarization on cloud height or temperature (Figure 6, Figure 8). Indeed most of these scatter plots show no noticable correlation. The descriptions of trends in the text are nonsensical. You might see more correlation if you allow the plots to display the variability of individual observations (or daily or hourly means, perhaps).

Although both ground-based and satellite lidar measurements are presented, there is very little attempt to make comparisons between them and no acknowledgement or explanation for cases where there are significant differences. For example, in figure 2, the satellite seems to see the cloud height as lower on average and the cloud temperature as significantly higher. Why? In Figure 6, the lidar ratios measured by the ground based and satellite measurements barely overlap. Why are there so many smaller ground-based values? Figure 8 shows an even more dramatic difference in the depolarization value. It is very concerning that the ground based measurements of cirrus

depolarization ratio (seasonal means) are often 5-10%. This seems very low. The only discussion is a brief speculation that it is due to supersaturated water clouds or horizontally oriented ice. If these explanations are true, do you understand why there no low values in the CALIOP dataset? Indeed there is no clear correspondence in any of the contour figures and nothing to even assure the readers that the ground based and satellite observations have anything in common at all.

Typos and other specific errors:

Line 100-101 on page 3, values of the horizontal resolution of both 5 km and 1000 m in adjacent sentences.

Line 140, page 4. December = November

Line 175, page 5. optical depth is not in units of km.

Line 146, page 4 (and other places in the manuscript) When giving ranges, give the smaller number first.

---

## Referee Comment (RC2) · Anonymous Referee #2 · 8 Aug 2016

General: The manuscript of Motty et al aims at the presentation of statistical properties of cirrus clouds over the tropical site of Gadanki . The paper reports cirrus observations performed within six years. The data are compared to CALIOP observations. For about 140-160 days (within 6 years!) cirrus properties are derived. All the data provide the impression that only random, snapshot-like observations are presented.

Because of the following reasons, the paper must be rejected:

Although of importance for lidar-derived cirrus studies the authors do not provide information whether or not the lidar is vertically pointing (yes it is vertically pointing, many data points are strongly corrupted by specular reflection, without stating that clearly), and what about multiple scattering effects: corrected or not corrected? Yes, the data

are not corrected, the reviewer noticed. So, all in all, the results are of rather limited use.

The spaceborne lidar CALIOP is not pointing nadir, so no specular reflection effects here? Comparison of results from zenith-pointing lidar and titled spaceborne lidar . . .., does that make any sense?

The extinction retrieval method is not explained? Just one sentence, to provide a reference for the retrieval, is simply not sufficient. Do you use backward and forward mode of the Fernald retrieval, to obtain the cirrus mean lidar ratio? Because the lidar ratios are not multiple-scattering-corrected, what is then the value of the data for further studies. . .?

Depolarisation characterization is simply not up-to-date. The molecular depolarization is defined by the interference filter width of the individual lidar. It is not stated whether the particle depolarization ratio or the volume depolarization ratio is used here, neither for the ground-based lidar nor for CALIOP .

Why are the depolarization results shown in Figure 7 different to those already shown in the publication of Motty et al. [2015]? In addition, in the current manuscript there is no reference given to the already published depolarization dataset of Motty et al. [2015].

All the correlations show almost no tendency, and are based on only 20-30 data points (measured within 6 years). Depolarization ratios are corrupted by specular reflection and lidar ratios, optical depth and extinction coefficients by multiple scattering effects. So, almost useless for further studies!

All the color plots are snapshots, and the results (ground-based versus spaceborne lidar) are sometimes rather different, because of the viewing geometry (from space the cirrus is in a distance of several 100 km, extremely large multiple scattering effects, from ground, cirrus is just 10-15 km away. . ., less strong, but still significant multiple

scattering), . . . and . . . zenith pointing at ground, near-nadir from space. . .).

Last point, although published (and thus this lidar-based method to estimate the vertical velocity can be used), I do not trust vertical (terminal) velocity observations based on just elastic backscatter lidars (i.e., if no vertically pointing Doppler lidar is used). All the structures (virga) you see with lidar at a given fixed site depends on the vertical profiles of horizontal wind velocity and wind direction at cirrus and virga height level, gravity waves have an impact, and of course sedimentation speed. So too many impacts to obtain the terminal velocity of ice crystals with sufficient accuracy.